# In Vivo Reversal of P-Glycoprotein-Mediated Drug Resistance in a Breast Cancer Xenograft and in Leukemia Models Using a Novel, Potent, and Nontoxic Epicatechin EC31

**DOI:** 10.3390/ijms24054377

**Published:** 2023-02-22

**Authors:** Wenqin Sun, Iris L. K. Wong, Helen Ka-Wai Law, Xiaochun Su, Terry C. F. Chan, Gege Sun, Xinqing Yang, Xingkai Wang, Tak Hang Chan, Shengbiao Wan, Larry M. C. Chow

**Affiliations:** 1State Key Laboratory of Chemical Biology and Drug Discovery, Department of Applied Biology and Chemical Technology, Hong Kong Polytechnic University, Hong Kong SAR, China; 2Department of Health Technology and Informatics, Hong Kong Polytechnic University, Hong Kong SAR, China; 3Laboratory for Marine Drugs and Bioproducts of Qingdao, National Laboratory for Marine Science and Technology, Key Laboratory of Marine Drugs, Ministry of Education, School of Medicine and Pharmacy, Ocean University of China, Qingdao 266003, China; 4Department of Chemistry, McGill University, Montreal, QC H3A 2K6, Canada

**Keywords:** multidrug resistance, MDR, P-glycoprotein, P-gp, epicatechin, EC, modulator

## Abstract

The modulation of P-glycoprotein (P-gp, ABCB1) can reverse multidrug resistance (MDR) and potentiate the efficacy of anticancer drugs. Tea polyphenols, such as epigallocatechin gallate (EGCG), have low P-gp-modulating activity, with an EC_50_ over 10 μM. In this study, we optimized a series of tea polyphenol derivatives and demonstrated that epicatechin **EC31** was a potent and nontoxic P-gp inhibitor. Its EC_50_ for reversing paclitaxel, doxorubicin, and vincristine resistance in three P-gp-overexpressing cell lines ranged from 37 to 249 nM. Mechanistic studies revealed that **EC31** restored intracellular drug accumulation by inhibiting P-gp-mediated drug efflux. It did not downregulate the plasma membrane P-gp level nor inhibit P-gp ATPase. It was not a transport substrate of P-gp. A pharmacokinetic study revealed that the intraperitoneal administration of 30 mg/kg of **EC31** could achieve a plasma concentration above its in vitro EC_50_ (94 nM) for more than 18 h. It did not affect the pharmacokinetic profile of coadministered paclitaxel. In the xenograft model of the P-gp-overexpressing LCC6MDR cell line, **EC31** reversed P-gp-mediated paclitaxel resistance and inhibited tumor growth by 27.4 to 36.1% (*p* < 0.001). Moreover, it also increased the intratumor paclitaxel level in the LCC6MDR xenograft by 6 fold (*p* < 0.001). In both murine leukemia P388ADR and human leukemia K562/P-gp mice models, the cotreatment of **EC31** and doxorubicin significantly prolonged the survival of the mice (*p* < 0.001 and *p* < 0.01) as compared to the doxorubicin alone group, respectively. Our results suggested that **EC31** was a promising candidate for further investigation on combination therapy for treating P-gp-overexpressing cancers.

## 1. Introduction

Multidrug resistance (MDR) is one of the major challenges in chemotherapy. The overexpression of ATP-binding cassette (ABC) transporters on the plasma membrane is one of the mechanisms for causing MDR in cancer. ABC transporters being overexpressed in naive or posttreatment tumors can limit the effectiveness of chemotherapeutic agents. They can actively efflux a broad range of structurally different cytotoxic drugs out of the cancer cells and, thus, reduce the intracellular drug concentrations [1,2]. P-glycoprotein (P-gp, MDR1 or ABCB1) is the best-characterized and most widely studied ABC transporter [3]. It is a homodimeric protein with two transmembrane domains (TMDs) and two cytosolic ATP nucleotide-binding domains (NBDs) [1]. The two TMDs form a large cavity for substrate binding and transport. The two NBDs are involved in ATP binding and hydrolysis to support the efflux [1,4]. P-gp overexpression has been correlated with poor clinical outcomes for breast cancer [5,6], sarcoma [7,8], and leukemia [9,10,11]. Many conventional anticancer drugs are P-gp substrates, including anthracyclines (doxorubicin (DOX) and daunorubicin), vinca alkaloids (vincristine (VCR) and vinblastine), colchicine, epipodophyllotoxins (etoposide and teniposide), paclitaxel (PTX), and imatinib [12]. One approach to tackling P-gp-mediated cancer MDR is to inhibit the transport activity of P-gp and to restore the anticancer drug concentration in the tumor [13,14]. 

First- and second-generation P-gp inhibitors are P-gp substrates, and they can compete with other substrates for binding sites, therefore acting as competitive inhibitors. These two generations of inhibitors include verapamil, cyclosporine, quinidine, reserpine, dexverapamil (R-isomer of verapamil), and PSC833 (a cyclosporine A analogue) [15,16]. They all failed in their clinical trials because of their low potency and high toxicity induced by unpredictable pharmacokinetic interactions with the anticancer drugs [17,18]. Third-generation modulators were developed through quantitative structure–activity relationship (QSAR) studies and high-throughput screening, including XR-9576 (tariquidar) [19], LY335979 (zosuquidar) [20,21], and elacridar [22]. They are noncompetitive P-gp inhibitors and have a high affinity for P-gp [23]. They bind at sites other than the substrate binding site of P-gp [24]. They are highly selective for P-gp, potent, and safe in in vitro studies, but some of them also failed in their clinical trials [20]. 

One factor for this failure resulted from the fact that the above-mentioned clinical trials did not stratify the patients based on the P-gp expression level in the tumor and the nonideal pharmacokinetic profiles of the modulators [18,25]. MDR can also be caused by other ABC transporters, such as breast cancer resistance protein (BCRP) [26] or multidrug-resistance-associated protein 1 (MRP1) [27], which may also result in the failure of the clinical trials of the third-generation modulators. The future development of inhibitors of ABC transporters should focus on potency, specificity, and safety. 

Epigallocatechin gallate (EGCG) is a natural polyphenol and is the most abundant catechin found in green tea. Its P-gp-modulating activity was first reported in 2002 [28]. EGCG at 50 µM chemosensitized the P-gp-overexpressing cell line CH^R^C5 cells to vinblastine and restored its IC_50_ to the wild-type level. EGCG was a potential P-gp modulator for reversing MDR in cancer. Its low potency, however, precluded it from further development.

To potentiate the P-gp-modulating activity of EGCG, we previously designed and synthesized a catechin library containing 78 members of methylated epigallocatechin (EGC), methylated gallocatechin (GC), methylated epicatechin (EC), and methylated catechin (C) [29,30]. A structure–activity relationship (SAR) study revealed that the replacement of all the -OH groups in the A, B, and D rings with -OMe groups could significantly improve its efficacy. In addition, the presence of a longer and more rigid oxycarbonylphenylcarbamoyl linker between ring D and C3 [30] was also important (Table 1). Four methylated catechin derivatives, (2R, 3R)-**EGC23**, (2R, 3S)-**GC51**, (2R, 3R)-**EC31**, and (2R, 3S)-**C25**, were the most potent inhibitors in reversing P-gp-mediated PTX resistance in the P-gp-overexpressing LCC6MDR cells. These 4 potent modulators (1 µM) reduced the IC_50_ of PTX by a relative fold (RF) of 41 to 85 (Table 1), which was significantly better than the parent compound, EGCG, with an RF of 1.2 at 10 µM [29,30]. Moreover, they were strong P-gp inhibitors but weak or nonexistent BCRP and MRP1 inhibitors (Table 1). In the present study, these four potent methylated catechin derivatives were further characterized by their mechanism, pharmacokinetics, toxicity, and in vivo antitumor activity. 

## 2. Results

### 2.1. Methylated Catechin Derivatives Modulated P-gp-Mediated MDR In Vitro

In our previous study, **EGC23**, **GC51**, **EC31**, and **C25** were found to be potent P-gp inhibitors in vitro [29,30]. Here, they were further characterized in vitro and in vivo. Human breast cancer cell lines LCC6 and LCC6MDR, murine leukemia cell lines P388 and P388ADR, and human leukemia cell lines K562 and K562/P-gp were used. LCC6, P388, and K562 were the sensitive parental cell lines. LCC6MDR and K562/P-gp were P-gp transfectants. P388ADR was established through a stepwise selection with Adriamycin. It was found that LCC6MDR, K562/P-gp, and P388ADR cells overexpressed more total or plasma membrane P-gp than their parental cell lines (Figure 1A–C). The IC_50_ of DOX, PTX, and VCR was 6.2- to 168-fold higher in the resistant cell lines (Figure 1D). In addition, P-gp (red fluorescence) was found to be localized on the cell surface of the P-gp-overexpressing cell lines but was not detected on that of the parental cell lines (Figure 1E–G).

We determined the effective concentration (EC_50_) of the four methylated catechin derivatives and EGCG in reversing P-gp-mediated drug resistance. EC_50_ was defined as the concentration at which the modulator can reduce the IC_50_ of an anticancer drug by half in the P-gp-overexpressing cell lines. EGCG displayed no P-gp-modulating activity, with an EC_50_ > 10,000 nM (Table 2). In contrast, the four methylated catechin derivatives had an EC_50_ of 93 to 385 nM for reversing DOX resistance, 82 to 496 nM for reversing PTX resistance, and 37 to 107 nM for reversing VCR resistance (Table 2). Overall, their EC_50_ values were at least 20- to 270-fold lower than that of EGCG, indicating that the O-methylation of all the rings and the rigid oxycarbonylphenylcarbamoyl and oxycarbonylvinyl linkers between ring D and C3 were crucial pharmacophores for modulating P-gp activity (Table 2). 

The P-gp-modulating activity of methylated catechin derivatives was further studied by measuring the EC_50_. EC_50_ was defined as the concentration of modulators at which it can reduce IC_50_ of anticancer drug of a cell line by 50%. Three P-gp-overexpressing cell lines were employed, including breast cancer cell line LCC6MDR, mouse leukemia cell line P388ADR, and human leukemia cell line K562/P-gp. The anticancer drugs tested included DOX, PTX, and VCR. All values were presented as mean ± standard error of mean. N = 3–6 independent experiments. 

### 2.2. Methylated Catechin Derivatives Increased DOX Accumulation by Inhibiting the Transport Activity of P-gp

DOX is a fluorescent P-gp substrate and can be used to study the function of P-gp. It was found that the accumulation of DOX in P-gp-overexpressing cell lines LCC6MDR, P388ADR, and K562/P-gp was 2.8-fold (*p* < 0.001), 4.3-fold (*p* < 0.001), and 3.0-fold (*p* < 0.001) lower than that in their parental cell lines, respectively (Figure 2A–C). When used at 0.5 or 1 µM, **EGC23**, **GC51**, **EC31**, and **C25** could restore DOX accumulation by 1.5 to 2.5 fold in LCC6MDR cells (Figure 2A), by 2.2 to 4.7 fold in P388ADR cells (Figure 1B), and by 1.6 to 2.8 fold in K562/P-gp cells, respectively (Figure 2C). In contrast, EGCG at 1 µM had no effect on DOX accumulation.

We investigated if methylated catechin derivatives could inhibit P-gp-mediated drug efflux in P-gp-overexpressing cells. P388ADR cells were preincubated with DOX and were then resuspended in DOX-free medium with or without 1 μM **EGC23**, **GC51**, **EC31**, and **C25**. Without a modulator, intracellularly, the DOX level was maintained at 80% and 11% after 180 min for P388 and P388ADR cells, respectively (Figure 2D). When 1 μM **EGC23**, **GC51**, **EC31,** and **C25** were included, the DOX efflux rate was significantly reduced. After 180 min of efflux, the intracellular DOX level could be maintained at 61% (*p* < 0.001), 59% (*p* < 0.01), 44 % (*p* < 0.001), and 37%, respectively (Figure 2D). The above results demonstrated that the reversal of DOX resistance by methylated catechin derivatives was due to an inhibition of P-gp-mediated drug efflux, leading to increased drug accumulation and, thus, restoring the drug sensitivity.

### 2.3. Methylated EC31 and C25 Did Not Inhibit P-gp ATPase Activity

**EC31** and **C25** did not affect the plasma membrane level of P-gp [29,30]. The efflux activity of P-gp is driven by ATP hydrolysis. Here, we investigated whether **EC31** and **C25** could affect P-gp ATPase activity. Verapamil, a well-known stimulator of P-gp ATPase activity, could increase P-gp ATPase activity in a dose-dependent manner, increasing the activity from 1.2 fold at 10 µM to 3.8 fold at 100 µM (Figure 3A). In contrast, **EC31** and **C25** were weak stimulators of P-gp ATPase, with up to a 2.1- to 1.6-fold increase when a 70 µM modulator was used (Figure 3A). EGCG could slightly increase P-gp ATPase activity by 1.4 fold at 10 µM and then decrease it to 0.7 fold at 70 µM (Figure 3A). These results suggested that the inhibition of the P-gp efflux function by **EC31** and **C25** was not due to the inhibition of ATPase activity.

### 2.4. Methylated EC31 Was Not a P-gp Transport Substrate

P-gp is a transporter with a wide substrate specificity. **EC31** is a weak stimulator of P-gp ATPase (Figure 3A). We investigated whether **EC31** was a substrate of P-gp by analyzing its intracellular retention level. It was found that **EC31** was retained in a similar level in both LCC6 and LCC6MDR cells at all the concentrations tested (0.1, 1, 10, and 50 µM) (Figure 3B). For PTX, a known P-gp substrate, LCC6 accumulated 2.3- and 3.4-fold (*p* < 0.001) more PTX than LCC6MDR cells accumulated when 1 or 5 µM PTX was used (Figure 3B). Similar observations were found in K562 and K562/P-gp cells (Figure 3C). In the **EC31** efflux assay, K562 and K562/P-gp displayed similar efflux rates. After 150 min of incubation, there was about 23% and 16% of **EC31** remaining in both cell lines (Figure 3D). These results suggested that **EC31** was not a transport substrate of P-gp. 

### 2.5. Pharmacokinetic (PK) Study of Methylated Catechin Derivatives and Their Effects on the PK of PTX in Mice

Prior to the in vivo efficacy experiment, a PK study of **EGC23**, **GC51**, **EC31**, and **C25** was performed using either intravenous (10 mg/kg i.v.) or intraperitoneal (30 mg/kg i.p.) injection routes (Figure 4A–D). The detailed PK profile was summarized in Figure 4E. With an i.p. injection at 30 mg/kg, the plasma levels of **GC51**, **EC31**, and **C25** exceeded their corresponding in vitro EC_50_ values for over 18 h, whereas **EGC23** lasted for 12 h, yielding a high bioavailability ranging from 63% to 82% after the dose normalization of AUC_i.p._ to AUC_i.v_. (Figure 4A–E).

P-gp was widely distributed in multiple tissues and organs, including the gut, brain, kidney, liver, and placenta [31]. Therefore, it can affect the pharmacological behavior of drugs by regulating drug absorption, distribution, and elimination. Here, the plasma level of PTX with or without the coadministration of **GC51**, **EC31**, or **C25** was determined up to 420 min after its administration (12 mg/kg i.v.) (Figure 4F). The AUC_0–420min_ of PTX alone was 400,972 ng-min/mL. The coadministration of **EC31** slightly increased the AUC_0–420min_ of PTX to 478,526 ng-min/mL, but it was without statistical significance (*p* > 0.05) (Figure 4F). **C25** or **GC51** at 30 mg/kg i.p. significantly increased the AUC_0–420min_ of PTX by 1.5 fold (** *p* < 0.01) and 1.6 fold (** *p* < 0.01), respectively (Figure 4F). **EC31** was selected for an in vivo efficacy study because of its high potency and specificity in vitro [30], high bioavailability (82%), long duration of keeping its plasma level above its in vitro EC_50_ (>18 h), and insignificant effect on the bioavailability of PTX.

### 2.6. **EC31** Alone and Its Combination with PTX Did Not Induce Toxicity in BALB/c Mice

A toxicity study of **EC31** (30 or 60 mg/kg i.p.) with or without the coadministration of PTX (12 mg/kg i.v.) was performed, and the injections were administered to BALB/c mice 12 times every other day. It was found that the repeated administration of **EC31** (30 mg/kg) alone did not cause any mortality or toxicity symptoms (Figure 5A). The repeated coadministration of **EC31** (either 30 or 60 mg/kg) and PTX induced a small body weight loss which was not greater than 15% for more than 3 consecutive days, and no toxicity symptoms were observed. The mice regained their body weight after the treatment was completed (Figure 5A). This result suggested that the coadministration of PTX with **EC31** was well tolerated by the BALB/c mice. Therefore, the i.p. administration of **EC31** at 30 or 60 mg/kg would be used in a subsequent efficacy study.

### 2.7. **EC31** Increased the Intratumor PTX Concentration and Reversed the P-gp-Mediated PTX Resistance in the LCC6MDR Tumor Xenograft Model

After optimizing the administration route and dosage of **EC31**, we investigated whether the coadministration of **EC31** could inhibit P-gp transport activity in the LCC6MDR xenograft, thereby enhancing the intratumor PTX level. When **EC31** was administered alone (30 mg/kg i.p.), it could be maintained inside the LCC6MDR xenograft above its in vitro EC_50_ (94 nM) for 420 min (Figure 5B), suggesting that **EC31** was stable in the plasma and could be continuously delivered to the tumor. When **EC31** (30 mg/kg i.p.) and PTX (12 mg/kg i.v.) were coadministered, **EC31** could increase the intratumor PTX level by 6 fold (*** *p* < 0.001) at 180 min as compared to PTX alone (Figure 5B). After 300 min, the intratumor level of **EC31** was reduced to 94 nM, and the intratumor level of PTX was the same with or without **EC31**, suggesting that the first 300 min postadministration were an effective period for **EC31** to increase the intratumor PTX concentration. This result suggested that **EC31** should be administered twice a day to maintain a sufficient intratumor level of coadministered PTX. 

According to our established xenograft model of LCC6MDR, the injection of PTX alone (12 mg/kg i.v.) could inhibit the tumor growth of LCC6 but not that of LCC6MDR [32]. Here, we tested whether **EC31** could reverse P-gp-mediated PTX resistance in the LCC6MDR xenograft model (Figure 5C). The treatment was given every other day 14 times (q2d x 14) from day 0 to day 26. There were four treatment groups, namely (1) the solvent control, (2) PTX alone (12 mg/kg i.v.), (3) **EC31** (30 mg/kg i.p. at −1 h) plus PTX (12 mg/kg i.v.) plus **EC31** (30 mg/kg i.p. at +5 h), and (4) **EC31** (60 mg/kg i.p. at −1 h) plus PTX (12 mg/kg i.v.). The solvent control (group 1) and PTX alone (group 2) groups exhibited similar tumor growth rates (Figure 5C). Both of the combination treatment groups (groups 3 and 4) displayed promising efficacies and significantly suppressed the tumor growth of LCC6MDR. On day 30, combination treatment group 3 resulted in a 1.6-fold (*** *p* < 0.001) and 1.5-fold (*** p* < 0.01) reduction in tumor volume and tumor weight as compared to PTX alone (12 mg/kg i.v.) (Figure 5D). Combination treatment group 4 resulted in a 1.4-fold (*** *p* < 0.001 and *** p* < 0.01) reduction in tumor volume and tumor weight as compared to PTX alone (12 mg/kg i.v.) (Figure 5D). The tumor doubling times in combination treatment groups 3 and 4 were 15.0 (* *p* < 0.05) and 12.4 days, which were significantly longer than that in the PTX alone group (11.1 days) (Figure 5D). No animal deaths were found in any of the four treatment groups (Figure 5D). These data suggested that **EC31** could increase the intratumor PTX level by inhibiting the transport activity of P-gp in the xenograft, thereby chemosensitizing the tumor cells to PTX.

### 2.8. **EC31** Reversed the DOX Resistance in the Murine Leukemia P388ADR and Human Leukemia K562/P-gp Models

We also tested the in vivo efficacy of **EC31** in reversing P-gp-mediated DOX resistance in the murine leukemia P388ADR model in B6D2F1 mice (Figure 6A). It was found that the untreated group possessed the shortest median survival of 13.0 days. DOX (3 mg/kg i.p.) alone prolonged the median survival to 16.5 days. The cotreatment of **EC31** (60 mg/kg i.p.) and DOX (3 mg/kg i.p.) extended the median survival to 21.5 days (Figure 6A), resulting in an increase in lifespan (ILS) of 30.3% (*** *p* < 0.001) compared to DOX alone (Figure 6A). The results suggested that **EC31** could potentially modulate P-gp-mediated DOX resistance in the murine leukemia P388ADR model and could prolong the lifespan. 

To determine the in vivo efficacy of **EC31** in reversing DOX resistance in the human leukemia K562/P-gp model, the survival time of leukemia-bearing NOD/SCID mice was monitored. The median survival time of the untreated group, the DOX alone (0.9 mg/kg i.v.) group, and the cotreatment (**EC31** 30mg/ kg i.p. + DOX 0.9 mg/kg i.v.) group was 29.0, 32.0, and 37.0 days, respectively (Figure 6B). The DOX alone group had a 10.3% ILS (*p* > 0.05) compared to the untreated group. The cotreatment group could prolong the animal survival by 5 days compared to the DOX alone group, representing a 15.6% ILS (** *p* < 0.01) (Figure 6B). This result suggested that the DOX alone treatment could not prolong the animal survival. **EC31** could potentiate DOX in treating the K562/P-gp leukemia-bearing mice and could, finally, prolong the animal survival.

## 3. Discussion

EGCG is the most abundant polyphenol in green tea. Its P-gp-modulating activity was first reported in 2002 [28]. EGCG at 50 µM chemosensitized P-gp-overexpressing cell line CH^R^C5 cells to vinblastine and lowered its IC_50_ to the wild-type level. Despite its potential to reverse MDR in cancer, its low potency limited it from further development. The modification of the OH groups to methoxy groups at the 3′ position of the B ring (EGCG-3′OMe) can improve the P-gp-modulating activity of EGCG by 2 fold [33].

We previously demonstrated that it was possible to modify EGCG to other novel catechin analogs to improve its P-gp-modulating activity. There were three important pharmacophores that were needed for the P-gp-modulating activity of EGCG, including (1) replacing all the –OH groups in the A, B, and D rings with -OMe groups, (2) conjugating the rigid oxycarbonylphenylcarbamoyl linker between ring D and C3, and (3) substituting the B ring with dimethoxylation rather than trimethoxylation [29,30]. Comparatively, stereochemistry had less of an effect on the P-gp-modulating activity of catechins [30]. EGCG displayed no P-gp-modulating activity even at 10 µM, with an RF of 1.2 (Table 1). After structural modification, (2R, 3R)-*cis*-methylated **EC31** and (2R, 3S)-*trans*-methylated **C25** were much more potent, with an RF of 69 to 85 at 1 µM (Table 1). They were nontoxic to the fibroblast cells, with an IC_50_ of >100 µM, and were highly selective for the P-gp transporter [29,30]. Here, we demonstrated that they could reverse drug resistance towards a panel of anticancer drugs, including DOX, PTX, and VCR, in three P-gp-overexpressing cell lines (EC_50_ = 93 to 260 nM for reversing DOX resistance, EC_50_ = 91 to 249 nM for reversing PTX resistance, and EC_50_ = 37 to 60 nM for reversing VCR resistance) (Table 2). They were at least 38- to 270-fold more potent than the parent compound EGCG. 

The mechanistic study demonstrated that the reversal of P-gp-mediated drug resistance by these methylated catechin derivatives was due to the inhibition of the efflux activity of P-gp (Figure 2D), restoring drug accumulation to a cytotoxic level (Figure 2A–C). They did not decrease the P-gp level at the plasma membrane [30] nor inhibit the P-gp ATPase activity (Figure 3A) to enhance drug retention. In the P-gp ATPase assay, **EC31** and **C25** were found to be weak stimulators of P-gp ATPase, with 1.6- to 2.0-fold stimulation at 50 µM. However, their stimulatory activities were not as strong as that of the well-known stimulator of P-gp ATPase, verapamil, which stimulated ATPase by about 3.2-fold (Figure 3A). Verapamil is a known P-gp transport substrate and worked as a competitive inhibitor to reverse MDR [34]. In the **EC31** accumulation study, a similar amount of intracellular **EC31** was detected in the cells with or without the overexpression of P-gp (Figure 3B,C), suggesting that **EC31** was not a transport substrate of P-gp and did not work as a competitive inhibitor to reverse P-gp-mediated drug resistance. Tariquidar is known to be a nonsubstrate of P-gp, yet it can act as a noncompetitive inhibitor and stimulate P-gp ATPase activity by 10 fold [35,36]. Another example is the P-gp antibody MRK16 which can bind to the epitope at the extracellular surface, thereby simultaneously blocking drug efflux and stimulating ATPase by 2 fold [37]. The binding of tariquidar or antibody MRK16 may lock P-gp in an outward-facing conformation and hold the two NBDs in close proximity, causing continuous ATP hydrolysis [23]. With such a locked conformation, P-gp would have a reduced affinity towards the substrates. Our results suggested that **EC31** was not a substrate of P-gp yet was able to stimulate P-gp ATPase activity, which is similar to tariquidar. One major drawback of competitive inhibitors is that a higher dosage is needed, which would cause unpredictable side effects or toxicity. The EC_50_ of **EC31** for reversing PTX resistance in LCC6MDR was 94 nM, which was about 4.7-fold lower than that of verapamil (EC_50_ = 446 nM) [29].

The pharmacokinetics of methylated catechin derivatives **EGC23**, **GC51**, **EC31**, and **C25** were evaluated in mice (Figure 4). **EC31** was selected for the in vivo efficacy study because it exhibited 82% plasma bioavailability after i.p. injection at 30 mg/kg (Figure 4C) and because its plasma level could be maintained above its in vitro EC_50_ (94 nM) for longer than 18 h (Figure 4C). High bioavailability in the plasma and the long residence time inside the body implied that **EC31** would be sufficiently exposed to the tumor for its activity. One of the obstacles for developing the P-gp modulators in clinic was the toxicity induced by the drug–drug interactions (DDI) between anticancer drugs and the modulators [17,18,25]. Here, the cotreatment of **EC31** and PTX was found to have no significant effect on the plasma PTX level in vivo (Figure 4F). This was consistent with the observation that **EC31** (30 or 60 mg/kg i.p.) together with PTX (12 mg/kg i.v.) in vivo did not lead to a significant body weight reduction or significant animal deaths when compared with PTX alone (Figure 5A).

The in vivo efficacy of **EC31** to reverse P-gp-mediated drug resistance was investigated in three animal cancer models, including a human breast cancer tumor xenograft LCC6MDR, a human leukemia K562/P-gp model, and a murine leukemia P388ADR model. Cotreatment (**EC31** 30 mg/kg i.p. + PTX 12 mg/kg i.v. + **EC31** 30 mg/kg i.p.) exhibited potent P-gp-inhibitory activity and restored the antitumor activity of PTX in the LCC6MDR xenograft (Figure 5C). This combination significantly reduced the tumor volume by 1.6 fold (*** *p* < 0.001) (Figure 5C,D) and the tumor weight by 1.5 fold (** *p* < 0.01) (Figure 5D) compared to PTX alone. Moreover, **EC31** at 30 mg/kg i.p. was demonstrated to significantly increase the intratumor PTX levels by 6 fold (*** *p* < 0.001) at 180 min postadministration (Figure 5B) as compared to PTX alone. In both the murine leukemia P388ADR and human leukemia K562/P-gp models, cotreatment (**EC31** + DOX) significantly prolonged the survival of the mice, with an ILS of 30.3% (*** *p* < 0.001) and 15.6% (** *p <* 0.01), as compared to the DOX alone group (Figure 6A,B), respectively.

In summary, **EC31** was a potent, specific, and nontoxic P-gp inhibitor. It significantly reversed P-gp-meditated drug resistance in vitro and in vivo in the human breast cancer xenograft and in the human leukemia and murine leukemia animal models. **EC31** inhibited the efflux activity of P-gp, restored intracellular or intratumor drug accumulation, and eventually chemosensitized the P-gp-overexpressing cells or xenograft and the leukemia cancers to the anticancer drug again. Therefore, **EC31** is a potential candidate for further investigation on combination therapy for treating P-gp-overexpressing cancers.

## 4. Materials and Methods

### 4.1. Catechin Analogues and Cell Lines

The synthesis of compounds **EGC23**, **GC51**, **EC31**, and **C25** was reported previously [29,30], and their purity was found to be over 98% by HPLC. Human breast cancer cell lines MDA435/LCC6 and MDA435/LCC6MDR were kindly provided by Dr. Robert Clarke (Georgetown University, Washington DC, USA). Human leukemia cell lines K562 and K562/P-gp were a generous gift from Prof. Kenneth To (the Chinese University of Hong Kong, Hong Kong SAR). Murine lymphoma cell lines P388 and P388ADR were generously provided by the National Cancer Institute (Bethesda, MD, USA).

### 4.2. Cell Cultures

MDA435/LCC6 and MDA435/LCC6MDR cells were cultured in complete DMEM medium containing 10% FBS, 100 U/mL of penicillin, and 100 μg/mL of streptomycin. P388, P388ADR, K562, and K562/P-gp cells were cultured in complete RPMI1640 medium with 10% FBS, 100 U/mL of penicillin, and 100 μg/mL of streptomycin. All cell lines were cultured at 37 °C and 5% CO_2_. LCC6 and LCC6MDR were adherent cells, whereas P388, P388ADR, K562, and K562/P-gp were suspension cells.

### 4.3. In Vitro Cell Proliferation Assay

About 6500 cells of LCC6MDR or 10,000 cells of P388ADR or K562/P-gp were seeded to each well of 96-well plate and were then incubated at 37 °C overnight. Next morning, the cells were cotreated with modulators (**EGC23**, **GC51**, **EC31**, or **C25**) at different concentrations (0, 62.5, 125, 250, 500, and 1000 nM) with different anticancer drugs, including paclitaxel (PTX) (0, 1.6, 5, 15, 44, 133, and 400 nM), doxorubicin (DOX) (0, 0.16, 0.5, 1.5, 4.4, 13, and 40 µM), or vincristine (VCR) (0, 0.4, 1.2, 3.7, 11, 33, and 100 nM). The 96-well plate was further incubated at 37 °C for 72 h. Cell viability was determined using CellTiter 96 AQ_ueous_ Assay (Promega, Madison, WI, USA, #G1111) as reported previously [38]. The % of cell survival, IC_50_ of anticancer drugs towards the cell line, and EC_50_ of modulators were analyzed using nonlinear regression in Prism 5.0.

### 4.4. DOX Accumulation and Efflux Assays

In the DOX accumulation assay, 5 × 10^5^ cells in complete media containing 20 μM DOX and different doses of modulator (0, 0.1, 0.5, or 1 μM) were added into an Eppendorf tube. They were subsequently incubated at 37 °C for 2.5 h with gentle shaking at 200 rpm. In the efflux assay, 5 × 10^5^ cells were firstly preincubated with 20 μM of DOX for 45 min at 37 °C. Then, the cells were washed, were resuspended in drug-free complete medium with or without 1 μM modulator, and were further incubated at 37 °C. At 0, 60, 120, and 180 min, the cells were collected. The intracellular DOX level was measured using a flow cytometer (BD Accuri^TM^ C6) as reported previously [30].

### 4.5. Determination of Total P-gp Expression Using Western Blot

About 1 × 10^6^ cells were lysed with 100 μL RIPA lysis buffer (Thermo Fisher Scientific, Waltham, MA, USA, # 89900) at 4 °C for 30 min. The lysed cells were spun at 14,000 rpm at 4 °C for 10 min, and the supernatant was saved.

About 20 μg of cell lysate was loaded on a 7.5% SDS-PAGE and was then electroblotted onto PVDF membrane (Millipore, Burlington, MA, USA, #IPVH00010). The membrane was firstly blocked with 5% nonfat dry milk in TBST buffer (0.05% Tween-20; 10 mM Tris-buffer, pH 7.5; and 150 mM NaCl) for 1 h at room temperature and was then incubated with 1:1000 primary antibody mouse anti-P-gp (Santa Cruz, Dallas, TX, USA, #SC-55510,) or 1:3000 mouse anti-β-actin (Santa Cruz, #SC-47778) at room temperature for 1 h. Subsequently, the membrane was incubated with 1:3000 secondary antibody goat antimouse-IgG-HRP (Santa Cruz, #SC-516102) for 1 h at room temperature. A chemiluminescent substrate (Millipore, WBKLS0500) was added to the membrane, and its signal was detected using ChemiDoc^TM^ Touch Gel Imaging System (Bio-Rad, Hercules, CA, USA, #1708370). In the Western blot membrane, total P-gp and β-actin expression in each lane was quantified with ImageJ software. Relative P-gp expression was an intensity ratio of P-gp band relative to the β-actin band.

### 4.6. Determination of Plasma Membrane P-gp Level Using Flow Cytometer

About 1 × 10^6^ cells were resuspended in 43 µL of FACS buffer (1% BSA and 1 mM ETDA in 1XPBS, pH7.4). A total of 2 µL of 1 µM vinblastine and 5 µL of PE labelled antihuman P-gp antibody (BD, Franklin Lakes, NJ, USA, #557003) were added to the cell suspension, and it was then incubated at 37 °C for 1 h. After staining, the level of P-gp-PE was determined as reported previously [30].

### 4.7. Determination of P-gp Cellular Localization Using Immunofluorescence Staining

About 1 × 10^5^ cells were seeded on a sterile round glass coverslip (12 mm in diameter) in a 24-well plate overnight. For suspension cells, the coverslip was pretreated with poly-L-lysine for cell adhesion. The cells were fixed with 4% paraformaldehyde for 15 min at room temperature and were subsequently permeabilized with 3% bovine serum albumin (BSA) and 0.1% triton-X100 in PBS for 30 min at room temperature. After PBS washing, the cells were incubated with 1:100 primary antibody of mouse anti-P-gp (Santa Cruz, #SC-55510) at 4 °C overnight. The cells were then incubated with 1:500 antimouse secondary antibody conjugated with Alexa Fluor 594 (Invitrogen, Waltham, MA, USA, #R37115) for 1 h at room temperature. Finally, the cells were counterstained with 1 μg/mL of DAPI (4′,6-diamidino-2-phenylindole) for 10 min at room temperature. The localization of P-gp in LCC6, LCC6MDR, P388, P388ADR, K562, and K562/P-gp cells was determined using a confocal microscope (Leica TCS SPC model no. DMi8).

### 4.8. P-gp ATPase Activity Assay

P-gp ATPase activity was measured with P-gp-Glo assay system (Promega, #V3601) with human P-gp membrane vesicles as reported previously [32]. The vesicles were incubated with or without 100 µM of sodium vanadate containing 0.1% DMSO, verapamil (10, 20, 50, and 100 µM), or methylated catechin derivatives (**EC31**, **C25**, or EGCG at 10, 50, and 70 µM). The reaction was initiated by adding MgATP and was then incubated at 37 °C for 1 h. After adding ATP detection reagent, the sample was incubated at room temperature for 20 min. The remaining ATP signal of luminescence was measured with a Clariostar microplate reader (BMG Labtech, Ortenberg, Germany). Vanadate-inhibitable P-gp ATPase activity was determined by calculating the luminescence signal in the samples with or without sodium vanadate.

### 4.9. UPLC-MS/MS Detection of Methylated Catechin Derivatives and PTX

Agilent 6460 Ultra-Performance Liquid Chromatography–Electrospray Ionization Triple Quadrupole mass spectrometer (UPLC-QQQ-MS) was used in determining the concentration of methylated catechin derivatives and PTX in the cell lysates, mice plasma, and tumor, respectively. Waters Acquity UPLC BEH C18 (1.7 μm, 2.1 × 50 mm) column was used as the stationary phase. Distilled H_2_O with 0.1% formic acid (solvent A) and acetonitrile with 0.1% formic acid (solvent B) were developed as mobile phase. The mobile phase was run at a flow rate of 0.3 mL/min. The linear gradient was set as follows: 20–45% solvent B at 0–2 min, 45–60% solvent B at 2–13 min, and 90% solvent B at 14–15 min. It then decreased to 20% over 1 min. The total run time was 16 min.

The gradient elution for PTX was run as follows: 10% solvent B at 0–1 min, 10–85% solvent B at 1–4 min, and 85% solvent B at 4–5 min. It then decreased to 10% solvent B over 1 min. A 90% solvent A and 10% solvent B was used for equilibration at 6–8 min. 

Mass spectrometers were operated in positive electrospray ionization (ESI+) mode with a supply gas and sheath gas temperature of 300 °C, 8 L/min of drying gas, 11 L/min of sheath gas flow, and 3.5 kV capillary voltage. To detect **EGC23**, **GC51**, **EC31**, and **C25**, the ion pairs used in MRM detection in triple quadruple MS (QQQ-MS) were set at 579.2 to 181.1, 708.3 to 181.1, 678.3 to 151.0, and 678.2 to 151.0, respectively.

### 4.10. Intracellular **EC31** Accumulation and Efflux

In the **EC31** accumulation assay, 5 × 10^5^ cells were incubated with different concentrations of **EC31** (0.1, 1, 10, or 50 µM) at 37 °C for 2.5 h. After incubation, the medium was removed, and cells were washed with cold PBS twice. To the cell pellet, 300 μL of acetonitrile was added. The cells were sonicated for 10 min at 50 Hz using Intertek-CD-3800(A) sonicator. The lysed cells were then spun at 14,000 rpm for 10 min at room temperature. The supernatant containing **EC31** was saved and was filtered with 0.45 µm syringe filter before UPLC-MS/MS analysis. For the efflux assay, the cells were preincubated with 1 μM **EC31** at 37 °C for 2.5 h. The cells were then washed with PBS, were resuspended in drug-free medium, and were further incubated at 37 °C. The cells were collected at 0, 30, 60, and 150 min, respectively. The intracellular **EC31** at different time points was analyzed as described in the Section 4.9.

### 4.11. Pharmacokinetic Study of Methylated Catechin Derivatives and PTX in Plasma

All animal experiments were conducted with approval from the Animal Subjects Ethics Sub-committee of the Hong Kong Polytechnic University and under the Animal (Control of Experiments) Ordinance Cap. 340 of the Hong Kong Government.

PTX stock (40 mg/mL) was prepared in 50% ethanol and 50% Cremophor EL. Before injection, stock solution was freshly diluted to 4 mg/mL with PBS (final formulation used was 5% ethanol, 5% Cremophor EL, and 90% PBS). Stock of methylated catechin derivatives at 30 mg/mL was prepared in 50% N-methyl-2-pyrrolidone (NMP) and 50% Cremophor EL. Before injection, the stock solution was freshly diluted to 3 mg/mL with PBS (final formulation used was 5% NMP, 5% Cremophor EL, and 90% PBS).

The effect of administration route (either intravenous or intraperitoneal) on the pharmacokinetics of **EGC23**, **GC51**, **EC31**, or **C25** alone was studied. In group 1, BALB/c mice were injected intravenously (i.v.) with 10 mg/kg of modulator, and their blood was collected at 10, 30, 60, 120, 240, and 480 min postadministration. In group 2, BALB/c mice were injected intraperitoneally (i.p.) with 30 mg/kg of modulator, and their blood was collected at 10, 30, 60, 180, 420, 840, and 1080 min postadministration.

The pharmacokinetics of PTX was studied with or without the coadministered modulator. PTX (12 mg/kg i.v.) alone was injected into the BALB/c mice. For the combination study, modulator (30 mg/kg i.p.) was injected 1 h before injecting PTX (12 mg/kg i.v.). The blood was collected at 10, 30, 60, 120, 240, and 420 min after PTX administration. Blood samples were spun at 14,000 rpm for 10 min at 4 °C to collect plasma supernatant. Plasma concentration of methylated catechin derivatives or PTX was determined through UPLC-QQQ-MS as described in Section 4.9. Pharmacokinetic parameters were analyzed with PK Solutions 2.0 (Summit Research Service, Ashland, OH, USA).

### 4.12. In Vivo Toxicity Study of **EC31** and Its Combination with PTX

The 6–8-week-old female BALB/c mice were divided into 5 groups (n = 5 mice in each group): (1) solvent control, (2) PTX at 12 mg/kg i.v., (3) cotreatment of **EC31** at 30 mg/kg i.p. and PTX at 12 mg/kg i.v., (4) cotreatment of **EC31** at 60 mg/kg i.p. and PTX at 12 mg/kg i.v., and (5) **EC31** at 30 mg/kg i.p., respectively. In the cotreatment groups, **EC31** was injected 1 h before PTX administration. The treatment was performed every two days 12 times (q2d × 12). During the study period, the behavior and the body weight of mice were monitored. 

After the last treatment on day 22, the body weight of mice was monitored for 7 days to assess delayed toxicity. The % of body weight loss was calculated as reported previously [32]. Over 15% body weight loss in 3 consecutive days, slowness in activity, and treatment-related death were all regarded as toxicological symptoms.

### 4.13. In Vivo Efficacy of **EC31** in Reversing PTX Resistance in LCC6MDR Tumor Xenograft Model

The LCC6MDR tumor xenograft model was studied as it was previously [32]. About 1 × 10^6^ cells of LCC6MDR cells were inoculated subcutaneously (s.c.) into the rear flank of a female BALB/c nude mouse (4–6 weeks old). When a tumor of 200–250 mm^3^ was formed, it was removed and cut into cubes of 1 mm^3^ and was xenografted s.c. into the rear flank of another female BALB/c nude mouse. Treatments started when the tumor reached 100 mm^3^. Mice were randomly divided into 4 groups with 8 mice in each group: (1) solvent of **EC31** (5% NMP + 5% Cremophor EL + 90% PBS i.p.) + solvent of PTX (5% Ethanol + 5% Cremophor EL + 90% PBS i.v.), (2) PTX alone (12 mg/kg i.v.) + solvent of **EC31** (i.p.), (3) cotreatment with **EC31** (30 mg/kg i.p.) being injected 1 h before PTX (12 mg/kg i.v.) and with **EC31** (30 mg/kg i.p.) being administered in another injection after 5 h postinjection of PTX, and (4) cotreatment with **EC31** (60 mg/kg i.p.) being injected 1 h before PTX (12 mg/kg i.v.) administration. Tumor growth was measured with an electronic caliper on the injection day, and the tumor volume was calculated as reported previously [32].

### 4.14. In Vivo PTX and **EC31** Accumulation in Tumor

LCC6MDR xenograft was established as described in Section 4.13. Treatment was given once to the mice when the tumor reached 150–300 mm^3^. There were three groups: (1) **EC31** (30 mg/kg i.p) alone, (2) PTX (12 mg/kg i.v.) alone, and (3) cotreatment with **EC31** (30 mg/kg i.p.) being injected 1 h prior to PTX (12 mg/kg i.v.) administration. Tumors were excised at 60, 180, 300, 420, and 540 min after PTX injection. Each cut tumor was homogenized with 3-fold volume of PBS. A total of 100 μL of homogenate was mixed with 300 μL of acetonitrile. The **EC31** and PTX levels in tumors were detected through UPLC-MS/MS as described in Section 4.9.

### 4.15. In Vivo Efficacy of **EC31** in Reversing DOX Resistance in Murine Leukemia P388ADR Model

About 1 × 10^6^ cells of P388ADR were i.p. inoculated in male B6D2F1 mice aged 5–7 weeks old. There were 3 groups: (1) control group with no treatment, (2) DOX (3 mg/kg i.p.) alone, and (3) cotreatment with **EC31** (60 mg/kg i.p.) being injected 1 h prior to DOX (3 mg/kg i.p.) administration. In group 2, DOX was i.p. injected on days 2, 6, 10, 14, and 18 only. In group 3, **EC31** was i.p. injected on days 1, 2, 5, 6, 9, 10, 13, 14, 17, and 18, and DOX was i.p. injected on days 2, 6, 10, 14, and 18. Before each treatment, 1.5 mg/mL of DOX was freshly prepared using PBS. Mice survival and body weight were monitored during the study period. 

### 4.16. In Vivo Efficacy of **EC31** in Reversing DOX Resistance in Human Leukemia K562/P-gp Model

NOD/SCID mice (4 weeks old, female) were irradiated at 1.2 Gy in the MultiRad 225 X-ray irradiation system (Precision) on day 0. Within 24 h, 1 × 10^7^ cells of K562/P-gp in 200 μL of PBS were i.v. injected into irradiated NOD/SCID mice. The mice were randomly distributed into 3 groups: (1) control group (no treatment), (2) DOX (0.9 mg/kg i.v.) alone, and (3) cotreatment with **EC31** (30 mg/kg i.p.) being injected 1 h prior to DOX (0.9 mg/kg i.v.) administration. **EC31** and DOX were injected on days 7, 11, 19, 23, and 27. Before each treatment, 0.2 mg/mL of DOX was freshly prepared using PBS. Mice survival and body weight were monitored during the study period.

## Figures and Tables

**Figure 1 ijms-24-04377-f001:**
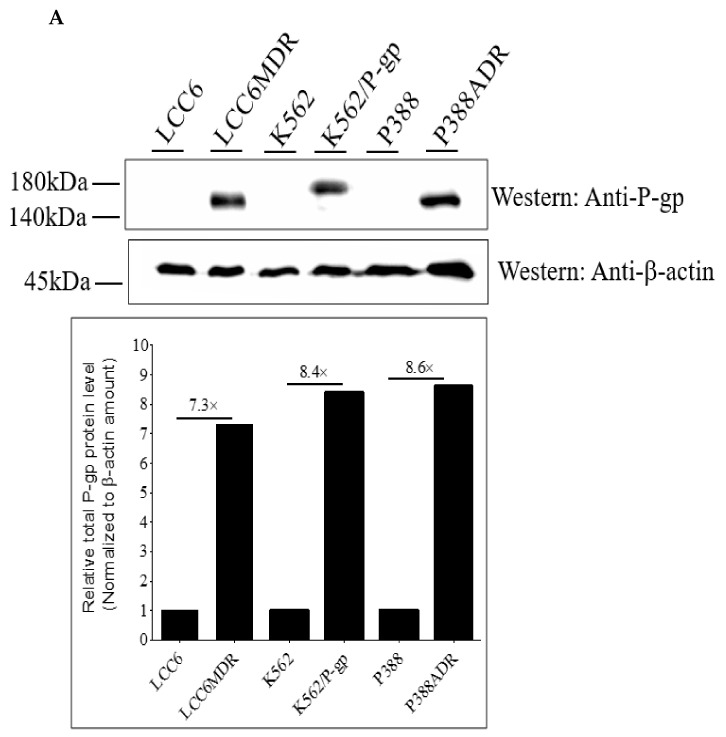
P-gp-overexpressing cell lines were more resistant to multiple anticancer drugs relative to their wild types and contained higher total or plasma membrane P-gp levels. (**A**) the total P-gp levels of three P-gp-overexpressing cell lines, including human breast cancer cell line LCC6MDR, mouse leukemia cell line P388ADR, and human leukemia cell line K562/P-gp, relative to their respective wild types, LCC6, P388, and K562, were studied through Western blotting using anti-P-gp antibodies. β-actin was the loading control. In the Western blot membrane, the total P-gp and β-actin expression in each lane was quantified with ImageJ software. The relative P-gp expression was an intensity ratio of P-gp band relative to the β-actin band. The relative P-gp expression of the resistant cell line relative to its respective wild type was presented as a bar chart. The plasma membrane P-gp levels of LCC6MDR and K562/P-gp relative to their wild types were determined through flow cytometry using antihuman P-gp-PE antibodies. (**B**) LCC6MDR and LCC6. (**C**) K562/P-gp and K562. (**D**) the drug resistance levels of three P-gp-overexpressing cell lines and their wild types were studied. The 3 pairs of cell lines were incubated with anticancer drugs paclitaxel (PTX), doxorubicin (DOX), and vincristine (VCR) for 3 days, and the % of cell survival was then determined. Their IC_50_ (nM) values were calculated using Prism 5.0. RF (relative fold) was a ratio of IC_50_ of anticancer drugs in a P-gp-overexpressing cell line relative to its respective wild type. N = 3 independent experiments. The IC_50_ values were presented as mean ± standard error of mean. The localization of P-gp in LCC6MDR, P388ADR, and K562/P-gp cell lines was also studied through immunofluorescence staining using primary anti-P-gp antibodies and antimouse Alexa Fluor 594-labelled IgG secondary antibodies. All cell lines were counterstained with DAPI. (**E**) LCC6 and LCC6MDR, (**F**) P388 and P338ADR, and (**G**) K562 and K562/P-gp. Bar = 8 µm. / = not determined.

**Figure 2 ijms-24-04377-f002:**
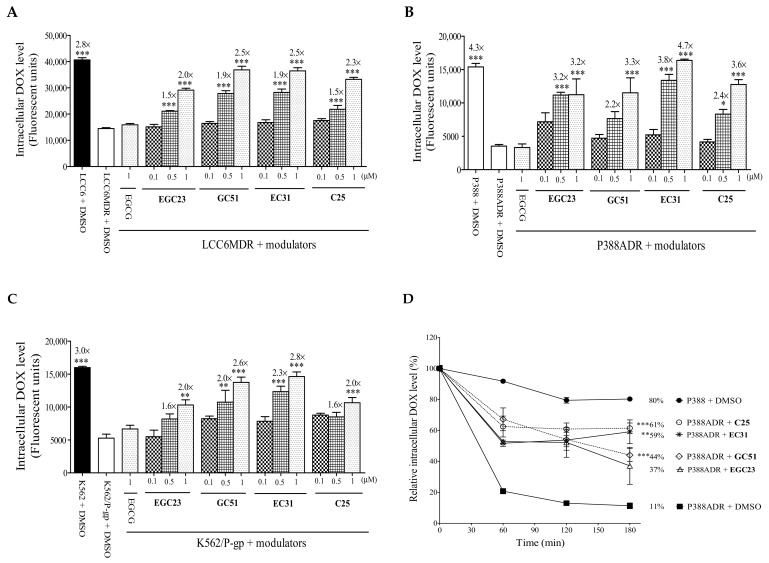
Effect of methylated catechin derivatives on intracellular DOX accumulation and DOX efflux in the P-gp-overexpressing cancer cells. The P-gp-overexpressing cells were coincubated with DOX (20 µM) and different concentrations of methylated catechin derivatives **EGC23**, **GC51**, **EC31**, and **C25** (0.1, 0.5, and 1 µM) for 150 min at 37 °C, respectively. Then, the cells were washed, and the intracellular DOX level was measured using flow cytometer. (**A**) LCC6MDR, (**B**) P388ADR, and (**C**) K562/P-gp. The parental cell lines LCC6, P388, and K562 were included in this assay. * represents *p* < 0.05, ** represents *p* < 0.01, and *** represents *p* < 0.001 relative to LCC6MDR, P388ADR, or K562/P-gp cells incubated with DMSO, respectively. (**D**) In the efflux study, P388ADR cells were firstly incubated with 20 µM DOX for 45 min. Then, the cells were washed and incubated with or without 1 µM **EGC23**, **GC51**, **EC31**, and **C25**. At 60, 120, and 180 min, the cells were harvested, and the intracellular DOX level was measured. At 180 min, ** represents *p* < 0.01, and *** represents *p* < 0.001, both being relative to P388ADR incubated with DMSO only. All values in this figure were presented as mean ± standard error of mean. N = 3 independent experiments. DMSO was used as a solvent control, and the highest amount used was 0.5%.

**Figure 3 ijms-24-04377-f003:**
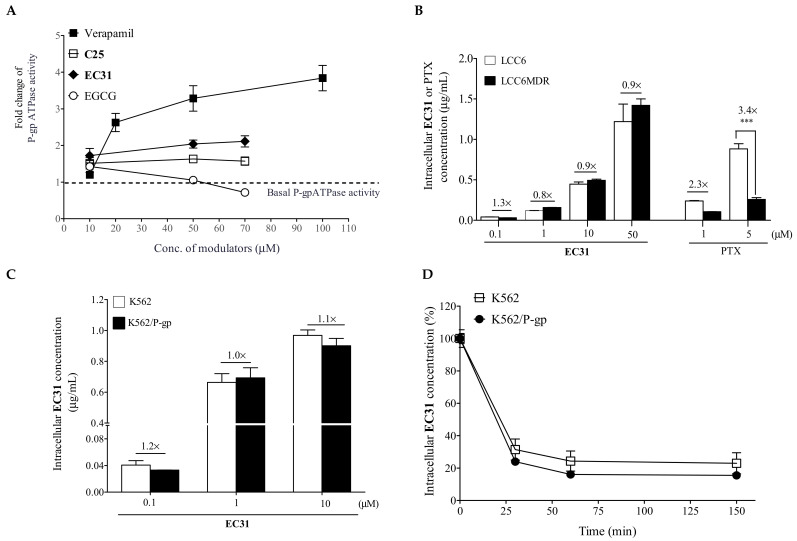
Mechanistic study of **EC31** in reversing P-gp-mediated drug resistance. (**A**) Effect of **EC31** on vanadate-sensitive P-gp ATPase was studied. **EC31**, **C25**, and EGCG at different concentrations (10, 50, and 70 µM) were incubated with recombinant human P-gp microsomes and MgATP with or without sodium vanadate. After stopping the reaction, the remaining ATP level was determined by measuring the luciferase-generated luminescent signal. Vanadate-inhibitable P-gp ATPase activity was determined by calculating the luminescence signal in the samples with or without sodium vanadate. Verapamil (10, 20, 50, and 100 µM) was used as a positive control for the stimulation of P-gp ATPase activity. P-gp ATPase activity caused by compounds was normalized to basal P-gp ATPase activity (human P-gp microsome treated with DMSO) and was presented as fold change. A dashed line at 1.0-fold change represented the basal P-gp ATPase activity. (**B**) Intracellular accumulation of **EC31** was studied in LCC6 and LCC6MDR cells and (**C**) in K562 and K562/P-gp cells. P-gp-overexpressing cells and their wild types were incubated with different concentrations of **EC31** (0.1, 1, 10, or 50 µM) for 150 min at 37 °C. The intracellular **EC31** level was determined through UPLC-MS/MS. The LCC6 and LCC6MDR cells were further incubated with the known P-gp substrate PTX (1 or 5 µM), which was a positive control. The data was analyzed through a one-way ANOVA. *** represents *p* < 0.001. (**D**) In the efflux study, K562 and K562/P-gp cells were firstly incubated with 1 µM **EC31** for 150 min at 37 °C. Then, the cells were washed and resuspended in drug-free medium. At 0, 30, 60, and 150 min, cells were harvested, and the intracellular **EC31** level was measured using UPLC-MS/MS. The values in this figure were presented as mean ± standard error of mean. N = 3 independent experiments.

**Figure 4 ijms-24-04377-f004:**
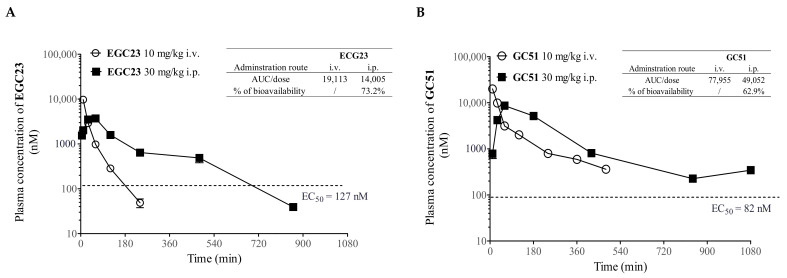
Pharmacokinetics of methylated catechin derivatives and their effects on the PK of PTX. BALB/c mice were i.v. injected with 10 mg/kg or i.p. injected with 30 mg/kg of modulators. At 10, 30, 60, 180, 420, 840, and 1080 min postadministration, animals were sacrificed to collect their blood, and then the plasma concentration of modulator was determined using UPLC-MS/MS. The % of bioavailability of each modulator injected through i.p. injection route was calculated by comparing it to its plasma level when injected via i.v. injection. (**A**) **EGC23**, (**B**) **GC51**, (**C**) **EC31**, and (**D**) **C25**. (**E**) Pharmacokinetic (PK) profiles of 4 potent methylated catechin derivatives were summarized. (**F**) The effect of modulators (**GC51**, **EC31**, or **C25**) on the PK of PTX was firstly determined through i.p. injection of modulator at 30 mg/kg 1 h prior to i.v. injection of 12 mg/kg PTX. At 10, 30, 60, 120, 240, and 420 min postadministration of PTX, animals were sacrificed. Plasma level of PTX was quantified through LC/MSMS. The dashed line indicates the corresponding EC_50_ of modulator in vitro for reversing PTX resistance. All values in this figure were presented as mean ± standard error of mean. There were 3 mice at each time point. All pharmacokinetic parameters were calculated with the pharmacokinetic software Summit^®^ PK solution. Two-tailed test was conducted for the AUC value of PTX alone and cotreatment. The dashed line in the figure represented the EC_50_ of each modulator used to reduce the IC_50_ of PTX by half in LCC6MDR cells. ** represents *p* < 0.01. N.S. = not significant. / = not applicable.

**Figure 5 ijms-24-04377-f005:**
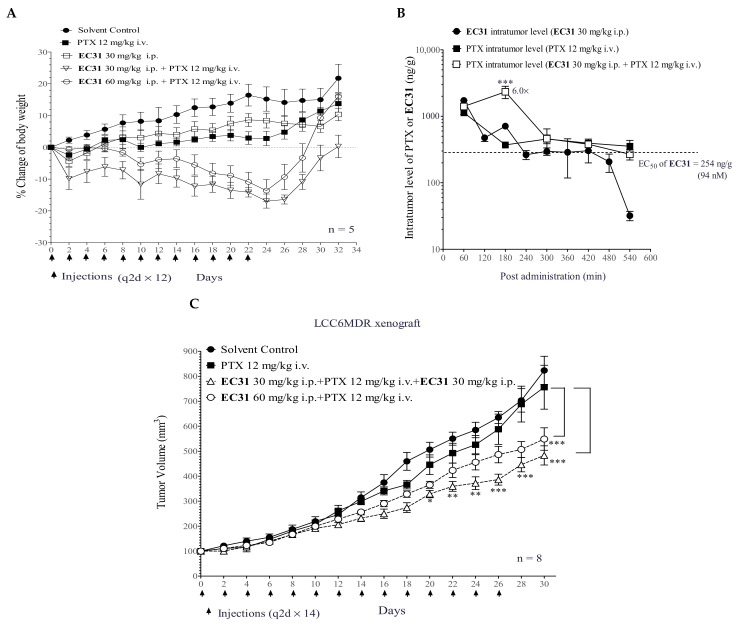
In vivo efficacy of **EC31** in reversing P-gp-mediated PTX resistance in LCC6MDR xenograft model. (**A**) In vivo toxicity study of **EC31** alone and of its combination with PTX was conducted. BALB/c mice were randomized into five groups with five mice in each group. Body weight of BALB/c mice during treatments was measured every two days. The treatment groups included (1) solvent control (5% NMP + 5% Cremophor EL + 90% PBS to dissolve **EC31** and 5% Ethanol + 5% Cremophor EL + 90% PBS to dissolve PTX), (2) PTX (12 mg/kg i.v.), (3) **EC31 (**30 mg/kg i.p.), (4) **EC31** (30 mg/kg i.p.) + PTX (12 mg/kg i.v.), or (5) **EC31** (60 mg/kg i.p.) + PTX (12 mg/kg i.v.). The administration was performed every other day 12 times (q2d x 12). The loss of body weight or appetite and treatment-related mortality were monitored and defined as toxicity. (**B**) BALB/c nude mice were subcutaneously xenografted with LCC6MDR. When the tumor reached 150–300 mm^3^, the mice were randomized and administered once with (1) **EC31** alone, (2) PTX alone, or (3) **EC31** + PTX. At different time points (60, 180, 300, 420, and 540 min) of postinjection, the mice were sacrificed. The tumors were taken out, and their weights were measured. Intratumor concentration of **EC31** or PTX was measured using UPLC-MS/MS, and the actual concentration of PTX or **EC31** was normalized to the tumor weight. The dashed line indicates the EC_50_ of **EC31** used for reducing IC_50_ of PTX by half in LCC6MDR cells. There were 3 to 5 mice at each time point. *** represents *p* < 0.001 relative to the PTX treatment alone at 180 min. (**C**) BALB/c nude mice were subcutaneously xenografted with LCC6MDR. The treatment groups included (1) solvent, (2) PTX (12 mg/kg i.v.) alone, (3) cotreatment with **EC31** (30 mg/kg i.p.) being injected 1 h prior to PTX (12 mg/kg i.v.) and with **EC31** (30 mg/kg i.p.) then being given in a second injection 5 h post-PTX injection, and (4) cotreatment with **EC31** (60 mg/kg i.p.) being injected 1 h prior to PTX injection (12 mg/kg i.v.). Tumor volume and body weight were monitored throughout the treatment period. There were 8 mice in each treatment group. Treatments were given as indicated by the arrows. (**D**) Effect of **EC31** on tumor volume, tumor weight, doubling time, and animal death in LCCMDR xenograft model. At the end of LCC6MDR xenograft efficacy study (day 30), all tumors were dissected. The tumor volume, tumor weight, and doubling time were measured. The number of animal deaths was recorded during the treatment period. Statistical analysis on tumor volumes of different treatment groups was done by performing two-way ANOVA with Dunnett’s multiple comparison test after treatment period. * represents *p* < 0.05, ** represents *p* < 0.01, and *** represents *p* < 0.001. All values were presented as mean ± standard error of mean.

**Figure 6 ijms-24-04377-f006:**
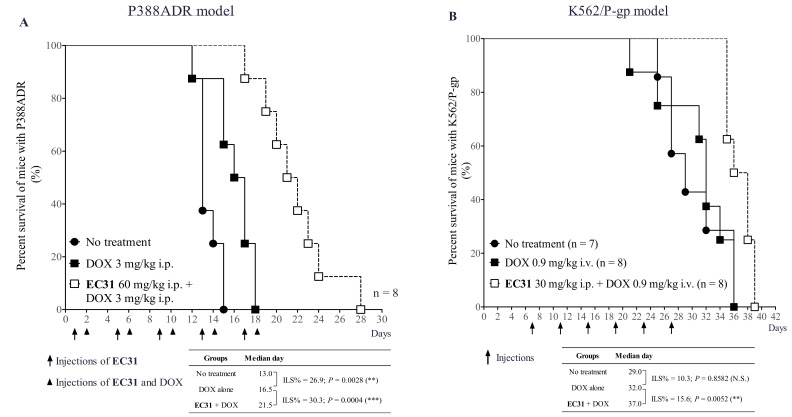
In vivo efficacy of **EC31** in reversing P-gp-mediated DOX resistance in P388ADR and K562/P-gp leukemia models. Irradiated BDF1 mice were i.p. inoculated with P388ADR cells, which caused mouse leukemia. Irradiated NOD/SCID mice were i.v. inoculated with K562/P-gp cells, which caused human leukemia. The mice were randomized and treated with (1) control treatment (no treatment), (2) DOX alone, and (3) **EC31** + DOX. Treatments were given as indicated by the arrows. The administration routes and dosages injected were indicated in the figure. The number of mice deaths in each treatment group was recorded every day. (**A**) The percent survival of mice infected with P388ADR. (**B**) The percent survival of mice infected with K562/P-gp. The mice median survival days and the increased lifespan (ILS) were analyzed with Log-rank (Mantel–COX) test in Prism 5.0. Student’s paired *t* test was conducted as indicated. ** represents *p* < 0.01, and *** represents *p* < 0.001. N.S. = not significant.

**Table 1 ijms-24-04377-t001:**
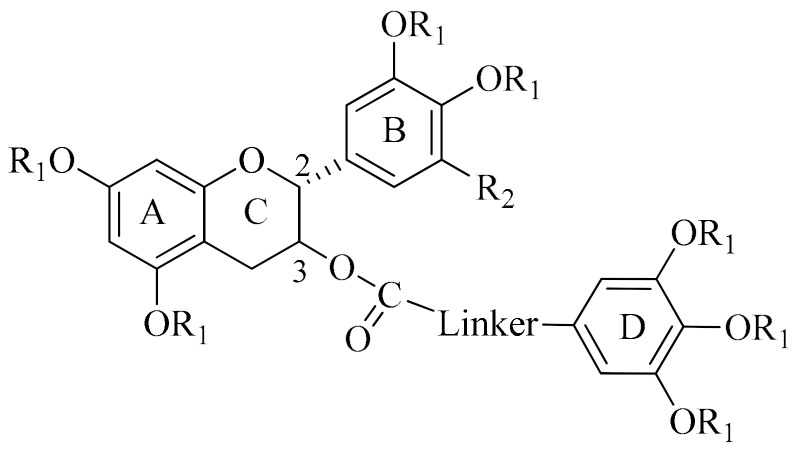
Chemical structure and selectivity of potent methylated catechin derivatives towards P-gp, BCRP, and MRP1 transporters.

Chemical Structures	Cpds at 1 μM
DMSO	EGCG (10 μM)	EGC23 ^a^	GC51 ^a^	EC31 ^b^	C25 ^b^
R_1_	/	H	Me	Me	Me	Me
R_2_	/	OH	OMe	OMe	H	H
Linker	/	/	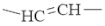	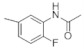	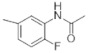	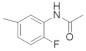
Position C_2_	/	R	R	R	R	R
Position C_3_	/	R	R	S	R	S
LCC6MDRP-gp-modulating activityMean IC_50_ of PTX (nM) [RF]	152.5 ^b^ [1.0]	122.6 ^a^ [1.2]	3.7 ^a^ [41.2]	2.9 ^a^ [52.6]	2.2 ^b^ [69.3]	1.8 ^b^ [84.7]
HEK293/R2BCRP-modulating activityMean IC_50_ of topotecan (nM) [RF]	295.6 ^b^ [1.0]	ND	357.0 ^a^ [ 0.8]	38.0 ^a^ [7.8]	100.8 ^b^ [2.9]	45.5 ^b^ [6.5]
2008MRP1MRP1-modulating activityMean IC_50_ of DOX (nM) [RF]	426.5 ^b^ [1.0]	ND	322.0 ^a^ [1.3]	141.0 ^a^ [ 3.0]	341.0 ^b^ [1.3]	353.7 ^b^ [1.2]

The selectivity of methylated catechin derivatives towards P-gp, BCRP, and MRP1 was studied previously using P-gp-overexpressing human breast cancer cell line LCC6MDR, BCRP-overexpressing human embryonic kidney cell line HEK293/R2, and MRP1-overexpressing human ovarian cancer cell line 2008/MRP1. ^a^ The chemical structure and ABC transporter selectivity of EGCG, **EGC23,** and **GC51** were published in 2015 [29]. ^b^ The chemical structure and ABC transporter selectivity of **EC31** and **C25** were published in 2021 (Reprinted/adapted with permission from ref. [30], copyright © 2021 by Elsevier. The mean IC_50_ of PTX, topotecan, and DOX with or without catechin derivatives is presented here. Relative fold (RF): IC_50_ of a drug of a cell line without modulator/IC_50_ of a drug of a cell line with modulator. ND = not determined. /: not applicable.

**Table 2 ijms-24-04377-t002:** EC_50_ (nM) of potent methylated catechin derivatives for reversing DOX, PTX, and VCR drug resistance in P-gp-overexpressing cell lines.

Cpds	EC_50_ (nM) for Reversing MDR					
	DOX Resistance			PTX Resistance		VCR Resistance
	LCC6MDR	P388ADR	K562/P-gp	LCC6MDR	P388ADR	K562
**EGC23**	244 ± 23	385 ± 16	165 ± 22	127 ± 30	168 ± 47	107 ± 6
**GC51**	233 ± 36	335 ± 15	129 ± 36	82 ± 14	496 ± 12	77 ± 7
**EC31**	159 ± 21	186 ± 31	93 ± 33	94 ± 21	249 ± 18	37 ± 10
**C25**	180 ± 18	260 ± 60	93 ± 21	91 ± 5	216 ± 10	60 ± 12
EGCG	>10,000	>10,000	>10,000	>10,000	>10,000	>10,000

## Data Availability

The data presented in this study are available on request from the corresponding author.

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
