# Peer review of "In Vivo Reversal of P-Glycoprotein-Mediated Drug Resistance in a Breast Cancer Xenograft and in Leukemia Models Using a Novel, Potent, and Nontoxic Epicatechin EC31"

_ijms, 2023, doi:10.3390/ijms24054377_

Round 1
Reviewer 1 Report (Previous Reviewer 2)
The figure was corrected and I am happy with the manuscript's quality.
Author Response
Thanks for your comments.
Reviewer 2 Report (Previous Reviewer 1)
Minor point:
Figure 1a - add densitometric assessment of P-gp expression.
Author Response
Figure 1a - add densitometric assessment of P-gp expression.
Reply: Thanks for your comments. The total P-gp expression level among the cell lines has been determined by the ImageJ software and summarized in a bar chart in Figure 1a (page 4).
This manuscript is a resubmission of an earlier submission. The following is a list of the peer review reports and author responses from that submission.
Round 1
Reviewer 1 Report
Major points
1. Section Introduction, first para second sentence: "The main cause of MDR was...", this is not true, at least not in clinical practice!
2. Section Introduction, should be rewritten to better explain this complicated issue. In addition, it would be appropriate to distinguish between competitive and non-competitive inhibitors of ABCB1.
3. Section MM should be rewritten. Do not describe trivial things, focus only on the essential ones so that your experiments can be repeated.
4. Section Results. The Figure 1 and Figure 2 are the same. Therefore, the text in section Results does not correspond to the Figure 1.
5. As you mentioned in Introduction, all aspects of ABCB1-mediated resistance depend on transporter expression level (Pharmacol Res. 2013 Jan;67(1):79-83, Chem Biol Interact. 2017 Aug 1;273:171-179.). Therefore, the expression levels of ABCB1 in used resistant cells should be demonstrated by western blot analysis.
6. It is unclear from your results and their interpretation the mechanism by which E31 inhibits ABCB1. On the one hand, EC31 significantly stimulates ATPase activity (Fig 2a) and at the same time is not a substrate of ABCB1 (Fig. 2B) - this is contradictory.
Reviewer 2 Report
The manuscript “In vivo reversal of P-glycoprotein-mediated drug resistance in breast cancer xenograft and leukemia models by a novel, potent and non-toxic epicatechin EC31” discusses the identification of a compound, epicatechin EC31, that is able to reverse P-gp-mediated multidrug resistance in vivo, ultimately improving the efficacy of anticancer drugs.
Despite many advances in anticancer treatment, chemotherapy is still a relevant strategy for treatment. However, multidrug resistance is still responsible for many deaths among cancer patients. The authors stated that despite continuing efforts to develop P-gp inhibitors to reverse multidrug resistance, many drugs still exhibit low potency or high toxicity.
The main contribution of the paper is the identification of EC31, a compound derived from epigallocatechin (abundant polyphenol in green tea), that was able to re-sensitize cells and mice to anticancer drugs (DOX and PTX), by inhibiting significantly P-gp activity through inhibition of its efflux activity. EC31 lack of toxicity might result partially from EC31 not being a P-gp substrate, and together with high and persistent plasma bioavailability, and increased animal survival, the authors suggest EC31 might be a promising compound for co-treatment with anticancer drugs.
The manuscript is clear, well structured, with relevant citations, with detailed methods descriptions. The images are easy to interpret and the authors conclusions are consistent with the data and arguments presented (with the exception of Figure 1 as stated below).
Section 3.2, from line 319 to 339, refers to DOX accumulation (Figure 1A, 1B and 1C0 and DOX efflux (Figure 1D) assays. Similarly, the legend for figure 1 reflects the same experiments. However, the Figure 1 panel displays a P-gp ATPase activity assay and 3 intracellular EC31 concentration assays. The same figure panel is presented in figure 2 (correctly).
